# One Session Effects of Knee Motion Visualization Using Immersive Virtual Reality in Patients with Hemophilic Arthropathy

**DOI:** 10.3390/jcm10204725

**Published:** 2021-10-14

**Authors:** Roberto Ucero-Lozano, Raúl Pérez-Llanes, José Antonio López-Pina, Rubén Cuesta-Barriuso

**Affiliations:** 1Department of Physiotherapy, European University of Madrid, 28670 Villaviciosa de Odón, Spain; ROBERTO.UCERO@universidadeuropea.es; 2Department of Physiotherapy, Catholic University San Antonio-UCAM, 30107 Murcia, Spain; rperez@ucam.edu; 3Department of Basic Phycology and Methodology, University of Murcia, 30100 Murcia, Spain; jlpina@um.es; 4Department of Physiotherapy, University of Murcia, 30100 Murcia, Spain; 5Royal Victoria Eugenia Foundation, 28029 Madrid, Spain

**Keywords:** hemophilia, quadriceps, knee arthropathy, motion visualization, electromyography

## Abstract

(1) Background: Hemophilic knee arthropathy is characterized by a loss of muscle mass and decreased strength of the quadriceps muscle. The visualization of movement aims to favor the recruitment of the motor system in the same premotor and parietal areas, as would happen with the active execution of the observed action. The aim was to evaluate changes in quadriceps activation in patients with hemophilic knee arthropathy following immersive VR visualization of knee extension movements. (2) Methods: We recruited 13 patients with severe hemophilia A and knee arthropathy. Patients underwent a 15 min session of immersive VR visualization of knee extension movements. The quadriceps muscle activation was evaluated by surface electromyography. (3) Results: After the intervention, there were no changes in the muscle activation of vastus medialis, vastus lateralis, or rectus femoris muscles. There was a large effect size of changes in rectus femoris muscle activation. Age and knee joint damage did not correlate with changes in quadriceps activation. Dominance, inhibitor development, and type of treatment were not related with post-intervention muscle activation. (4) Conclusions: A session of immersive VR visualization of knee extension movement does not modify quadriceps muscle activation. A specific protocol for patients with hemophilic knee arthropathy may be effective in improving the activation of the rectus femoris muscle.

## 1. Introduction

Hemophilia is a congenital coagulopathy characterized by a deficiency of a clotting factor. Depending on the missing factor, there are two types: hemophilia A (FVIII deficiency) or hemophilia B (FIX deficiency). Based on the percentage of the missing blood clotting factor, several hemophilia phenotypes are found: severe (<1% FVIII/FIX), moderate (1–5%) and mild (5–40%) [1].

Although hemophilia is a hematological disease, it typically evolves with the development of musculoskeletal bleeding, either spontaneously or triggered by minimal trauma. Most bleeding events are intra-articular (hemarthrosis) and mainly in the lower limbs, due to pressure and loads on the knees and ankles [2,3]. These bleeding events, recurring from early childhood, lead to degenerative and chronic joint deterioration (hemophilic arthropathy) [4]. The regular administration of clotting factor concentrates (prophylactic treatment) is the most effective way of controlling hemarthrosis and hemophilic arthropathy [5]. The development of antibodies against the treatment with clotting factor concentrates (inhibitors) is the main clinical complication in these patients and increases the rate of musculoskeletal sequelae.

Among other clinical symptoms, hemophilic arthropathy presents a reduced range of motion, biomechanical and perceptual alterations, and atrophy of the periarticular muscles [6]. Recurrent hemarthrosis and eventual joint deterioration, either more or less severe, induce poor activation of the quadriceps muscle [6]. Such muscular atrophy is similarly observed in patients with knee arthritis, where significant quadriceps weakness and atrophy has been described [7]. The reduced activation is shown by the lower electromyographic activity in the quadriceps, associated with reflex muscle inhibition, and may appear in the absence of structural damage, inflammation, or pain. The intra-articular infusion of 10 mL of sterile saline solution in healthy subjects can cause this reflex muscle inhibition [7,8]. This phenomenon is known as arthrogenic muscle inhibition [9].

Therapy using movement observation is a movement representation technique [10] which is neurophysiologically-based on mirror neurons and their activation when viewing the performance of a given action or one that is similar [11]. While watching a specific action, a recruitment of the motor system is induced for the same premotor and parietal areas that would occur with the active performance of the action displayed [11]. This occurs when the movement displayed is part of the motor scheme of the watching individual. When such movement is not part of the individual’s motor scheme, the action is broken down into more basic motor actions that activate the corresponding brain areas, thanks to the mirror mechanism [11]. This phenomenon can be used to elicit neuroplastic changes.

The goal of movement observation interventions is to visualize movements performed by a subject without injury or pain [10]. These interventions have shown their effectiveness in the treatment of acute and chronic pain in people with knee and hip arthroplasty [12] or chronic neck pain [13] Similarly, these interventions can improve the learning of motor tasks [10].

Chronic pain conditions such as phantom limb pain or complex regional syndrome are associated with changes and reorganization of the patients’ sensory and motor cortical networks. Movement observation aims to restore the integrity of neural processing in the sensory–motor cortex of these patients. Similarly, in patients with neurological and orthopedic pathologies such as stroke, Parkinson’s, and after knee or hip arthroplasty, an improvement in muscle performance of the agonist muscles, relative to the activity watched, was noted [14].

Movement observation can be used in different ways: from a first-person perspective or a third-person perspective, and with non-immersive video or immersive video where the subject feels present in the action [15]. Movement observation is more effective when the video to be viewed is immersive and recorded from a first-person perspective [15,16]. A video is recorded from a first-person perspective in which the observer is in the middle of the action. The use of highly realistic digital avatars promotes brain activation, produced by the viewing of the subject [17]. Dai et al. [18] noted how the degree of muscle activation is directly proportional to the brain activation of the primary and secondary motor areas, and associative cortex.

The aim of this observational study was to assess changes in the quadriceps muscle activation in a population of patients with hemophilic knee arthropathy following an immersive virtual reality visualization of knee extension movements.

## 2. Materials and Methods

### 2.1. Study Design

A prospective observational study to assess changes in the quadriceps muscle activation in patients with severe hemophilia A and hemophilic knee arthropathy was performed. The study period was from May to June 2021.

### 2.2. Local Approvals

One of the study researchers informed the patient associations about the selection criteria and goals set for the study. After gathering the potential participants, they were informed of study risks and benefits and provided with an information sheet with the most relevant data. All participants signed the Informed Consent Document.

The Clinical Research Ethics Committee of the Virgen de la Arrixaca University Hospital approved the study (2020-2-9-HCUVA). The study was conducted in accordance with the Declaration of Helsinki. The data included in this study were part of a larger research project on the safety and efficacy of visualization of movement in patients with hemophilic arthropathy (www.clinicaltrials.gov (accessed on 27 July 2021) ID: NCT04549402).

### 2.3. Participants

The sample size was calculated before or after the research was performed. The sample size was calculated using the statistical package G*Power (version 3.1.9.2; Heinrich-Heine-Universität Düsseldorf, Düsseldorf, Germany). Assuming a mean effect size (d = 0.80), with an alpha level (type I error) of 0.05 and a statistical power of 80% (1 − β = 0.80), a sample size of 12 patients with hemophilic knee arthropathy was estimated. Patients were recruited from three Spanish regions (Madrid, Murcia, and Malaga), through the Spanish Federation of Hemophilia.

The inclusion criteria were: patients diagnosed with severe hemophilia A; aged 18 years or over; hemophilic knee arthropathy (more than 3 points on the Hemophilia Joint Health Score) [19]; and to have signed the informed consent document. The excluded patients were those who had developed hemarthrosis in the previous three months, had visual problems that made it difficult to view movement on the mobile application, had difficulty in understanding the instructions for assessment, and had ankyloses of the knee hindering contraction of the quadriceps muscles. The development of antibodies to FVIII concentrates or the type of pharmacological treatment (prophylactic or on demand) were not included in the selection criteria.

### 2.4. Outcome Measures

Before the experimental session, the main clinical variables (type of treatment, knee joint condition, development of antibodies against clotting factor concentrates or inhibitors) and anthropometric variables (weight and height) of the patients recruited in the study were collected. A physiotherapist blinded to the study objectives performed all assessments. The primary variable was muscle activation of the vastus medialis, vastus lateralis, and rectus femoris of the quadriceps.

Muscle activation of the quadriceps was evaluated with surface electromyography (surface EMG model; Shimmer Sensing, Dublin, Ireland). Electrodes were placed according to the European recommendations for the use of surface electromyography [20]. Bipolar rectangular silver/silver chloride (Ag/AgCl) electrodes were used, measuring 28 × 44 mm (Ambu^®^ WhiteSensor 4200 model) with a 46 mm^2^ measurement area, 2 cm apart [21]. For the patients to reach their maximum muscle strength, the rater provided the same verbal stimulus to motivate all the patients with each contraction.

The skin was prepared according to the SENIAM project recommendations (surface EMG for non-invasive muscle assessment), by shaving and cleaning the area with alcohol, which was allowed to evaporate before placing the electrode [20].

Electrodes were placed following SENIAM recommendations [21]. To evaluate rectus femoris activation, the electrode was placed at 50% of the distance of the path that runs from the line between the anterosuperior iliac spine and the upper pole of the patella. For the measurement of the vastus lateralis activation, the electrode was placed at 2/3 of the distance of the line running from the anterosuperior iliac spine to the lateral face of the patella. Lastly, it was placed at 80% of the distance between the line from the anterosuperior iliac spine and the articular midline of the knee, in front of the anterior edge of the medial ligament to record vastus medialis activation. The reference electrode (ground) was placed on the lateral edge of the ankle (external malleolus). A baseline measurement was performed, and another after viewing the movement asking the patient to perform an isometric contraction of the quadriceps. To do this, the patient was placed in supine position with 75° hip flexion and knee flexion. A pressure dynamometer was placed perpendicular to the leg being evaluated, just above the lateral malleolus, maintaining 75° knee flexion. The patient was asked to conduct two 5 s maximum isometric contractions, with a 30 s break in between, against the dynamometer held by the evaluator. This protocol was based on the system described by Skou et al. [22]. However, patients were evaluated in supine position due to functional limitations in the lower limbs, typical in patients with hemophilic knee arthropathy.

### 2.5. sEMG Analysis

The reliable and validated surface electromyography (sEMG) mDurance^®^ system (mDurance Solutions SL, Granada, Spain) was used to record muscle activity during a functional task (ICC = 0.916; 95%CI = 0.831–0.958) [23]. The muscles assessed were the rectus femoris, vastus lateralis, and vastus medialis of the quadriceps. Data were obtained for both limbs.

The mDurance^®^ system (mDurance Solutions SL, Granada, Spain) consists of three parts: (a) a Shimmer3 EMG unit (Realtime Technologies Ltd., Dublin, Ireland), which is a bipolar surface electromyography sensor for the acquisition of muscle activity—each Shimmer3 has two channels, with a 1024 Hz sampling rate and applies an 8.4 Hz bandwidth, while the EMG signal resolution is 24 bits, with an overall amplification of 100–10,000 *v*/*v* [23]; (b) the mDurance Android application, which receives data from Shimmer3 and sends it to a cloud service [23]; (c) the mDurance cloud, service where data are stored, filtered and analyzed [23].

The raw data were processed and filtered using a fourth-order Butterworth bandpass filter with a 20–450 Hz cut-off frequency. The signal was smoothed using a window size of 0.025 s root mean square (RMS) and 0.0125 s overlap between windows [23]. The main variable recorded for muscle activity was the mean RMS, expressed in microV, of the middle third of the isometric contraction. The start and end of the signal were identified using a threshold method, and this was verified visually afterwards.

A threshold method was used to automatically identify the beginning and end of each repetition. For complete security, the visual inspection of all the points detected was performed by a member of our research group. The mid third portion of each repetition was used for variable extraction. Fast Fourier Transform analysis was performed. The variables extracted from this portion of the signal were the mean RMS and the median frequency. The values of all repetitions from the same subject and test were an average.

### 2.6. Intervention

The intervention consisted of a movement visualization by the reproduction of an immersive 180° video on a smartphone and viewed using virtual reality glasses (3D VR glasses with remote control; Q-MAX model 5802) [16]. In total, 150 flexion–extension movements of the knees were observed with a frequency of 10 repetitions per minute, based on the functional need for this movement in patients with hemophilic knee arthropathy. This video was filmed using an actor with full range of motion and without knee pathology. The intervention was performed with the patient sitting, resting both feet, and being fully relaxed. The patient did not have to perform any knee movements during the intervention. The intervention lasted 15 min. The patient was able to move his head freely but focused his attention on the movement of the body segment.

### 2.7. Statistical Analysis

Statistical analysis was performed with the SPSS statistical package (Version 21.0; IBM Corp., Armonk, NY, USA). The Shapiro–Wilk test was used to analyze whether sample distribution followed the normality criteria. A descriptive analysis was performed using the main statistics of central tendency (median) and dispersion (interquartile range). Changes after the experimental test were obtained through a paired samples t-test. The effect size was calculated using the Cohen’s D means difference formula [24]. The effect size was interpreted as small (d > 0.2), medium (d > 0.5), or large (d > 0.8). The correlation between changes in quadriceps activation and age and joint health was determined using the Pearson’s correlation coefficient. Differences in muscle activation were analyzed using a one-way ANOVA, based on type of treatment, dominance, and inhibitor development. The data were considered statistically significant when *p* < 0.05.

## 3. Results

### 3.1. Descriptive Analysis

The median age of the patients enrolled in the study was 37 (IR: 10) years, with a mean of 84.3 kg (IR: 14.65) in weight and 173 cm (IR: 8.5) in height. All patients had a medical diagnosis of hemophilia A with a severe phenotype (<1% FVIII). The median knee joint damage at the time of the study was 12 points (DT: 5.5) on the Hemophilia Joint Health Score. Most patients received prophylactic treatment with clotting factor concentrates (84.6%), and only 30.8% of patients had developed antibodies to the factor concentrates. Table 1 shows the descriptive statistics of the sample.

### 3.2. Changes after Intervention

When comparing the two evaluations performed in the study, no differences were found in the root mean square (*p* > 0.05). The effect size of the change in the rectus femoris muscle showed very high values (d = 0.89). Table 2 shows the central tendency and dispersion values of the analyses, the changes, and the calculation of the effect size.

When comparing the median frequency in both evaluations, differences were found in the vastus medialis (t = 2.62; *p* < 0.05). Table 3 shows the mean difference values of the median frequency and the changes.

### 3.3. Correlation Analysis

When comparing changes with the dichotomous qualitative variables (type of treatment, dominance, and development of inhibitors), we found no significant differences (*p* > 0.05) between muscle changes and type of treatment, dominance, and inhibitor development. The comparison analysis results are shown in Table 4. Figure 1 shows the changes in root mean square after movement visualization.

No correlation was found in the correlation analysis between the study muscle variables and the quantitative variables (age and knee joint damage). Table 5 shows the correlation analysis.

## 4. Discussion

The aim of this observational study was to assess changes in quadriceps muscle activation in a population of patients with hemophilic knee arthropathy following an immersive VR visualization of knee extension movements. After the viewing, no changes were found in vastus medialis, vastus lateralis, or rectus femoris activation.

Activation of the rectus femoris during knee extension, observed by electromyography, was greater than that experienced by the vastus medialis and vastus lateralis during isokinetic contractions [25]. The intervention comprised watching a video showing knee extension movements. The action was performed actively, freely, and slowly. This movement is visually similar to an isokinetic contraction.

Poortvliet et al. [26] reported a relationship between cortical and muscular activation in strength tasks and postural control. Watching the action favors the recruitment of the motor system in the same premotor and parietal areas as with the active execution [11]. Therefore, visualizing a free low-speed knee extension movement can generate greater cortical activation of the areas referring to the rectus femoris and, therefore, improved training thereof.

The effect of movement observation on the stimulation of the cerebral and cerebellar areas as part of the motor response, and its impact in terms of muscle activation response, has been addressed [27,28]. These changes can be induced by transcranial stimulation causing electromyographic muscle changes in healthy subjects, in people with Parkinson’s, and essential tremor. Brain stimulation was induced in our study by immersive VR visualization of movement.

The median frequency is defined as the frequency that divides the power density spectrum in two regions with the same amount of power [29]. Median frequency variation (VMF) shows neuromuscular fatigue during the various repetitions in the vastus medialis [21]. According to our study, this value showed statistically significant changes between the two assessments. These results are consistent with those reported by Tarata et al. [30], who reported a drop after the beginning of the task, showing how the central intervention affects fatigue. Similarly, Calatayud et al. [21] reported significant differences in low-load tasks. Three types of fatigue are described: central fatigue, neuromuscular junction fatigue, and muscle fatigue [31]. Based on the characteristics of the intervention carried out in our study, it is likely that the vastus medialis fatigue is of central origin.

We noted how joint damage and age did not correlate with changes in muscle activation. The study patients’ advanced knee joint damage can predict the efficacy of observation with movement in patients with hemophilic knee arthropathy. Similarly, the absence of differences in muscle activation depending on the type of treatment and the development of inhibitors allows us to be optimistic about the effectiveness of a specific protocol lasting several weeks, regardless of the therapeutic regimen and the development of antibodies to clotting factor concentrates.

The absence of significant differences in dominance in patients with hemophilia is not consistent with what has been disclosed by other studies [32,33]. This lack of differences based on the dominant side may be due to the unilateral observation of the action carried out. However, the video watched by patients with hemophilia showed bilateral knee extension movements, which may favor the activation of both sides, and the absence of bilateral differences.

Movement visualization has shown modifications in cortical activation and other brain areas by functional magnetic resonance imaging [34]. In patients with Parkinson’s disease, an increased recruitment of motor and fronto-parietal regions has been observed, manifesting in changes in blood oxygenation level-dependent contrast associated with the performance of the tasks that were assessed on a pixel-by-pixel basis, using the general linear model and the theory of Gaussian field. Although these changes in blood oxygenation were not evaluated in our study, it would be necessary to evaluate these modifications in cortical activation with movement visualization in patients with hemophilic arthropathy.

Obhi et al. [35] observed an increase in electromyographic muscle activity in healthy subjects exposed to movement visualization after a single session. This change was due to feedforward mechanisms prior to movement. Although in our study we did not observe significant changes in electromyographic activity, future studies should continue to study the electromyographic effect of movement visualization in patients with hemophilic arthropathy with muscle deficits.

### 4.1. Limitations of the Study

A limitation of our study may be the sample size. The sample size calculation is for a small effect size (0.30). Due to the low sample size of patients, the authors propose to subsequently develop a randomized clinical trial with a larger sample, using VR immersive visualization in patients with hemophilia.

The inclusion of patients with severe hemophilia A and knee arthropathy precludes extrapolation of the study results. The inclusion of patients with advanced joint damage may limit the effect of the intervention.

This study reflects the results of a one-time intervention in which immediate results are measured. The research was performed on only one session of VR immersive visualization. Since interventions are not periodic and there is no follow-up evaluation, it is not possible to establish the effectiveness of an immersive VR visualization of knee extension movements in patients with hemophilic arthropathy.

### 4.2. Relevance to Clinical Practice

The improvement of the quadriceps muscle activation in people with hemophilic knee arthropathy is a therapeutic goal in this population. The restricted range of knee motion and the chronic pain suffered by these patients require specific, safe, and effective therapy, promoting specific muscle activation.

On the other hand, this intervention is very safe, offering a valid option in patients with hemophilia, regardless of their therapeutic regimen and phenotype. Moreover, access to this technology is widely available.

The cost of virtual reality glasses needed for this study is very low, and the immersive video can be watched on the patient’s own smartphone, regardless of the model (Android^®^ or IPhone^®^). Easy access to the intervention implies a democratization of the treatment, as it can be readily used at home at a low cost.

### 4.3. Recommendations for Future Research

The development of randomized clinical studies implementing a periodic schedule of immersive VR visualization of knee extension movements in patients with hemophilic arthropathy could confirm the results. The recruitment of a larger sample size and the evaluation of clinical variables of functionality and chronic pain would allow us to confirm the safety and efficacy of immersive VR visualization of movement in patients with hemophilia and hemophilic arthropathy. Similarly, the assessment of muscle strength using dynamometry could indicate whether the administration of an immersive virtual reality visualization protocol can produce changes in the generation of muscle force, related to changes in muscle activation.

## 5. Conclusions

Immersive VR visualization of knee extension movements produces no change in quadriceps muscle activation in patients with hemophilic knee arthropathy. Immersive VR visualization of knee extension movements has the potential to activate the rectus femoris muscle based on the results of the large effect size. Age, dominance, or clinical status of the hemophilia patient does not promote changes in muscle activation following immersive VR visualization of knee movements. Randomized clinical studies could confirm the efficacy of an intervention by immersive VR visualization of knee extension movements in patients with hemophilic arthropathy.

## Figures and Tables

**Figure 1 jcm-10-04725-f001:**
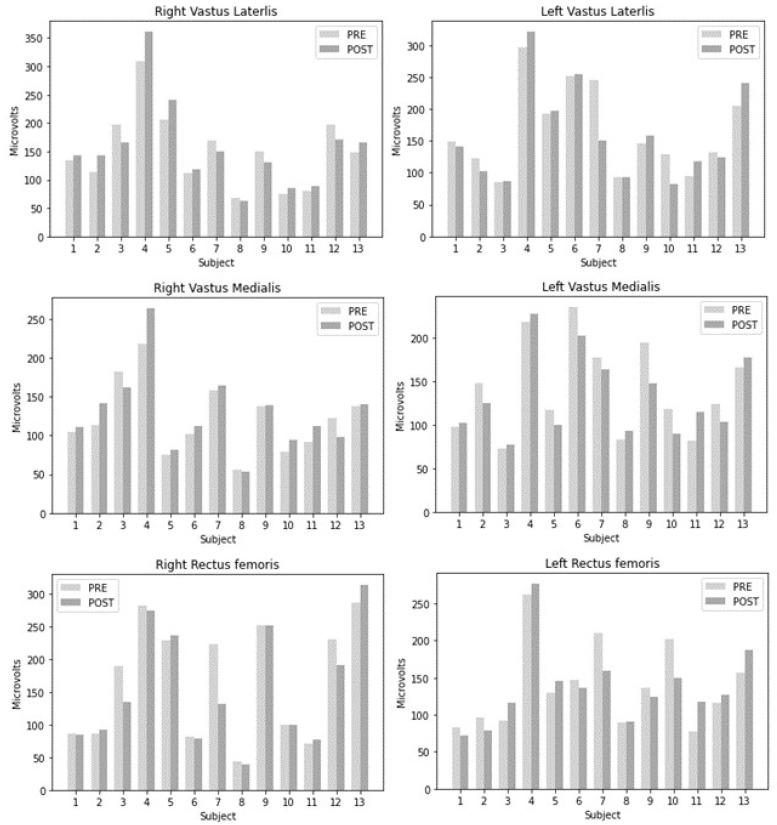
Changes in root mean square after intervention.

**Table 1 jcm-10-04725-t001:** Descriptive characteristics of the patients.

Variables	Md (IR)
Age (years)	37 (10)
Weight (kg)	84.30 (14.65)
Height (cm)	173.00 (8.5)
Knee joint damage (0–20)	12.00 (5.5)
	*n* (%)
Type of treatment (On demand/Prophylactic)	2/11 (15.4/84.6)
Development of inhibitor (No/Yes)	9/4 (69.2/30.8)

Md: median; IR: interquartile range.

**Table 2 jcm-10-04725-t002:** Means (standard deviations) of and changes in physical measurements evaluated in both assessments in root mean square.

Muscle	T0	T1	Sig.	CI95%	ES
Vastus lateralis	157.49 (66.25)	157.51 (73.87)	0.99	−12.14; 12.10	−0.01
Vastus medialis	130.98 (49.73)	130.54 (48.53)	0.27	−5.58; 18.69	0.01
Rectus femoris	145.80 (72.52)	152.36 (74.78)	0.91	−8.31; 9.19	0.89

T0: outcome measures before visualization; T1: outcome measures after visualization; Sig.: significance; CI95%: 95% confidence interval; ES: effect size.

**Table 3 jcm-10-04725-t003:** Mean differences (standard deviations) of and changes in physical measurements evaluated in both assessments in median frequency.

Muscle	MD (SD)	Sig.	CI95%
Vastus lateralis	−0.25 (9.67)	0.89	−4.15; 3.65
Vastus medialis	1.98 (3.86)	0.01	0.42; 3.54
Rectus femoris	0.03 (7.69)	0.98	−3.07; 3.14

MD: mean difference; SD: standard deviation; Sig.: significance; CI95%: 95% confidence interval.

**Table 4 jcm-10-04725-t004:** Changes according to the degree of hemophilia severity, type of treatment, and development of inhibitors.

Variables	Muscle	t (Sig.)	95%CI
Type of treatment	Vastus lateralis	0.58 (0.56)	−24.42 to 43.82
	Vastus medialis	1.73 (0.09)	−3.72 to 43.05
	Rectus femoris	1.10 (0.27)	−15.55 to 51.57
Dominance	Vastus lateralis	−1.21 (0.23)	(−27.64; 7.11)
	Vastus medialis	−0.55 (0.58)	(−31.25; 18.03)
	Rectus femoris	0.60 (0.55)	(−17.39; 31.88)
Development of inhibitors	Vastus lateralis	0.42 (0.67)	−21.23; 32.30
	Vastus medialis	1.02 (0.31)	−9.56; 28.40
	Rectus femoris	0.20 (0.84)	−24.23; 29.51

t: Levene statistics; Sig.: significance.

**Table 5 jcm-10-04725-t005:** Correlation between the observed difference in means and quantitative variables.

Muscle	Age		Knee Joint Damage	
	r	Sig.	r	Sig.
Vastus lateralis	0.11	0.57	−0.04	0.81
Vastus medialis	0.083	0.68	−0.177	0.38
Rectus femoris	0.14	0.49	0.07	0.73

r: Pearson’s correlation; Sig.: significance.

## Data Availability

Not applicable.

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
