# Peer review of "One Session Effects of Knee Motion Visualization Using Immersive Virtual Reality in Patients with Hemophilic Arthropathy"

_jcm, 2021, doi:10.3390/jcm10204725_

Round 1
Reviewer 1 Report
This study investigated the activation of the quadriceps muscle by VAR in hemophilia patients. I was interested in studying the effects of VAR on hemophilia patients who need patronizing treatment of the knee joint in physical therapy in rehabilitation. This study shows the potential of VAR intervention in hemophilia patients.
I have a few comments on the manuscript. Please consider making changes or corrections. I believe will improve the readability of the paper.
・This paper mainly conveys the effect of virtual reality intervention in hemophiliacs by mentioning the effect size of rectus femoris muscle activation, although there is no significant difference in the RMS of sEMG before and after the intervention. I assume that the author is taking a positive view of the possibility of what is negative data. However, when he does not acknowledge a significant difference in activation, he is limiting the scope of statistical interpretation. Therefore, I believe an emphasis on significance is needed more than it is now.
・Please indicate whether the results of this study will be specific to hemophilia patients. If it is possible to compare the results with those of healthy subjects, please add a note.
・Did you observe a 15-minute video of the patient's own knee joint flexion and extension exercises? Please consider adding this information.
・Please consider adding a note that FFT analysis was performed.
・Is the activation of the rectus femoris muscle alone a meaningful effect? After showing that there is an effect on performance in hemophiliacs, isn't it desirable to show the contents of this study as an explanation of the mechanism?
・The reference in VMF to fatigue is a description of what can be derived from the intervention method. However, wouldn't it be difficult to argue if the subject had free knee joint movement during the viewing?
・Line88 Please add the full spelling of the abbreviation "VR".
Best regards.
Author Response
Review Report (Reviewer 1)
This study investigated the activation of the quadriceps muscle by VAR in hemophilia patients. I was interested in studying the effects of VAR on hemophilia patients who need patronizing treatment of the knee joint in physical therapy in rehabilitation. This study shows the potential of VAR intervention in hemophilia patients. I have a few comments on the manuscript. Please consider making changes or corrections. I believe will improve the readability of the paper.
-
This paper mainly conveys the effect of virtual reality intervention in hemophiliacs by mentioning the effect size of rectus femoris muscle activation, although there is no significant difference in the RMS of sEMG before and after the intervention. I assume that the author is taking a positive view of the possibility of what is negative data. However, when he does not acknowledge a significant difference in activation, he is limiting the scope of statistical interpretation. Therefore, I believe an emphasis on significance is needed more than it is now. The significance value (p <.05) alone offers no more value than the fact that there are changes between two evaluations. On many occasions, when the sample size is low, the type I error characteristic of significance offers us a non-significant value. In order to know the true power of the results, the effect size was calculated. This statistic offers us, beyond the sample size, the power of the results obtained. However, to avoid confusion for the reader, and as indicated by the reviewer, we have redrafted the Discussion section concentrating the discussion on the results of significance.
-
Please indicate whether the results of this study will be specific to hemophilia patients. If it is possible to compare the results with those of healthy subjects, please add a note. In this case, the results are specific to patients with hemophilia since the entire sample were patients with hemophilia. This same experiment could be done with healthy subjects and see on one side if there is an effect and on the other to compare it with our sample. However, this study is specifically designed in patients with hemophilia because our target population as a researcher team is hemophilia. Therefore, we wanted to see that modifications occurred from the first moment in these subjects. For us it is important to know this due to muscle inhibition that patients with hemophilic arthropathy suffer, as it can be an effective and safe way to activate the musculature.
-
Did you observe a 15-minute video of the patient's own knee joint flexion and extension exercises? Please consider adding this information. No, it is a video previously recorded in immersive 180º video recorded in the first person. In that video you can see the full flexion and extension movement. The fact that the video is immersive makes the subject assimilate what he sees as his own, being the equivalent of the rubber hand experiment or mirror therapy treatments in which the patient feels the perceived illusion as real. It is a brain "trick" effect. We have modified the sentence to make it easier for the reader to understand.
-
Please consider adding a note that FFT analysis was performed. As the reviewer points out, it has been included
-
Is the activation of the rectus femoris muscle alone a meaningful effect? After showing that there is an effect on performance in hemophiliacs, isn't it desirable to show the contents of this study as an explanation of the mechanism? That the rectus femoris muscle is the only muscle that presents a high effect is significant in relation to the movement seen in the training session. This muscle is the one that will work the most during the entire flexion-extension movement against gravity with a seated patient if we actively perform this movement. The mechanism of action is based on the neurological activation that is caused by seeing the movement and that ultimately forms part of the effector system.
-
The reference in VMF to fatigue is a description of what can be derived from the intervention method. However, wouldn't it be difficult to argue if the subject had free knee joint movement during the viewing? Totally agree with the reviewer, if we understand fatigue as something that only involves the muscle fiber. The subject in this intervention does not perform any movement, however, the fatigue does not have to be only muscular. In this case, the perceived fatigue would be neurological, since during the visualization of movement there is neurological activity in the same way that it would occur in an active movement. What would influence muscle contraction
-
Line88 Please add the full spelling of the abbreviation "VR". As the reviewer points out, we have included the complete word of the VR abbreviation
Reviewer 2 Report
Major concerns
The authors should reconsider their paper because pilot studies have other outcomes and goals. Perhaps the authors should consider if this is a pilot study how the research protocol fitted for a larger randomized controlled trial. The pilot study is also used to identify the validity of a further randomized controlled trial. Please check the next two references for pilot study research design and outcomes:
- In, Junyong. “Introduction of a pilot study.” Korean journal of anesthesiology vol. 70,6 (2017): 601-605. doi:10.4097/kjae.2017.70.6.601
- Thabane, L., Ma, J., Chu, R. et al. A tutorial on pilot studies: the what, why and how. BMC Med Res Methodol 10, 1 (2010). https://doi.org/10.1186/1471-2288-10-1
Also regarding the paper, the experiment was performed on one session of Immersive VR visualisation, therefore, the results, conclusions and title can not be extrapolated.
The sample size calculation is for a small effect size (0.30), also, the research was performed on only one session of VR immersive visualisation. These two are major limitations of the research. The authors should emphasize these aspects in the limitations section. Or discuss them regarding the pilot study goals.
There are no further specific data regarding the treatment received by the enrolled patients, and this is another major limitation that could interfere with the results.
No data regarding the muscle strength of the quadriceps muscles, although a dynamometer was also used in the assessment. Could the strength interfere with motor neuron fibre recrutation and upper nervous system plasticity response? Also, the authors should emphasize that neuroplasticity is developed in time, 15 minutes are not enough for the conclusions presented.
Since the research is a pilot study, the discussion section should be adjusted according to pilot study goals. Also the conclusion section.
Minor concerns
Why the authors used median and interquartile range in the descriptive characteristics of the patients? ( and in Table 1- delete "at baseline" since only one session of VR was performed).
Is not clear if the patients only visualised the movement or also moved the segment. Please specify clearly.
Since the paired samples T-test was used to identify the changes after the experimental test, it seems that all data were normally distributed, but in the results sections, in table 4 are shown some Mann Whitney test and in not clear why.
Line 229- which were the dichotomous variables and what kind of test the authors used?
Line 123, and Table 4 results- is not clear what "inhibitors" means.
In the discussion section, the authors should emphasize that even a large effect size was observed in rectus femoris activation, the results were not significant, and the p value was not even close to 0.5. Also, the authors should know that the effect size can be biased in sample sizes smaller than 50 participants.
How many repetitions of knee extensions had the participants visualised in this experimental research?
Author Response
Review Report (Reviewer 2)
Major concerns
-
The authors should reconsider their paper because pilot studies have other outcomes and goals. Perhaps the authors should consider if this is a pilot study how the research protocol fitted for a larger randomized controlled trial. The pilot study is also used to identify the validity of a further randomized controlled trial. In accordance with the comments of the reviewer, the study design has been reformulated to avoid unnecessary confusion.
-
Also regarding the paper, the experiment was performed on one session of Immersive VR visualisation, therefore, the results, conclusions and title can not be extrapolated. Strongly agree with the reviewer that the results cannot be extrapolated in the long term. The objective of the study, modified based on the study design, is to evaluate whether the visualization of movement causes immediate changes in muscle recruitment. Undoubtedly, the evaluation of its effects in the medium or long term must be analyzed in an RCT. In accordance with the comments of the reviewer, the title, objectives, Methods and Results of the study have been reformulated to avoid confusion for the reader.
-
The sample size calculation is for a small effect size (0.30), also, the research was performed on only one session of VR immersive visualisation. These two are major limitations of the research. The authors should emphasize these aspects in the limitations section. Or discuss them regarding the pilot study goals. According to the author, the main limitation of the study is the size of the sample. This aspect has been included in the study limitations section for quick identification. In the same way, the need to perform RCTs that show the results of VR immersive visualization in patients with hemophilia has been included.
-
There are no further specific data regarding the treatment received by the enrolled patients, and this is another major limitation that could interfere with the results. As the reviewer points out, we should add the exact dosage. Information has been included regarding the number of knee flexion-extension movements (150) at a cadence of 10 repetitions per minute.
-
No data regarding the muscle strength of the quadriceps muscles, although a dynamometer was also used in the assessment. Could the strength interfere with motor neuron fibre recrutation and upper nervous system plasticity response? Also, the authors should emphasize that neuroplasticity is developed in time, 15 minutes are not enough for the conclusions presented. The aim of using a dynamometer was to always request a maximum contraction of 5 seconds from the patient. The dynamometer with an auditory stimulus at 5 seconds facilitates its performance for the patient. In this case the force data were not recorded because due to the mechanism of action in no case would we have adaptations in force. Strength is one of the characteristics of muscle contraction, but in this case, how that muscles are activated is evaluated. If we go to the other extreme and think of high-level sports subjects with great strength development, we can find subjects that do not have a good activation pattern. We couldn't agree more with the reviewer that neuroplasticity develops over time. Therefore, these changes were not expected to be maintained over time if no more sessions were carried out since learning (in this case neuromotor) requires time. However, this work indicates the short-term effect of neuronal activation caused by seeing a movement; how you could improve that recruiting almost immediately. This finding could be very useful clinically to improve motor control in patients with hemophilic arthropathy before exposing them to an active phase.
-
Since the research is a pilot study, the discussion section should be adjusted according to pilot study goals. Also the conclusion section. As indicated by the reviewer 1, we have redrafted the Discussion and Conclusions sections, concentrating the discussion on the results of significance, according to a cohort design.
Minor concerns
-
Why the authors used median and interquartile range in the descriptive characteristics of the patients? (and in Table 1- delete "at baseline" since only one session of VR was performed). These values of central tendency (median) and dispersion (interquartile range) were used because the sample was small. Given a larger effect size, we would use other statistics (mean and standard deviation). Even so, if the reviewer prefers the inclusion of mean and standard deviation, we would make the change. According to the reviewer, being a study with an immediate pre-post design, the word baseline should not appear
-
Is not clear if the patients only visualised the movement or also moved the segment. Please specify clearly. According to the reviewer's comment, to avoid confusion for the reader we have added the following sentence: "Patient have not to do any knee movements during the intervention"
-
Since the paired samples T-test was used to identify the changes after the experimental test, it seems that all data were normally distributed, but in the results sections, in table 4 are shown some Mann Whitney test and in not clear why. Indeed, the reviewer's doubt is due to an error in the transcription of table 4. Its involvement in the results table has been corrected.
-
Line 229- which were the dichotomous variables and what kind of test the authors used? The dichotomous variables are: Type of treatment, Dominance and Development of inhibitors. In the same paragraph it is indicated that the results are observed in table 4 where they are indicated. To facilitate their location, they have been written in the text, in the Results
-
Line 123, and Table 4 results- is not clear what "inhibitors" means. One of the main complications in the treatment of patients with hemophilia is the development of antibodies against clotting factor concentrates (inhibitors). This information has been included in the Introduction section to facilitate the reader's understanding of the use of the Inhibitor variable in the results.
-
In the discussion section, the authors should emphasize that even a large effect size was observed in rectus femoris activation, the results were not significant, and the p value was not even close to 0.5. Also, the authors should know that the effect size can be biased in sample sizes smaller than 50 participants. The significance value (p <.05) alone offers no more value than the fact that there are changes between two evaluations. On many occasions, when the sample size is low, the type I error characteristic of significance offers us a non-significant value. In order to know the true power of the results, the effect size was calculated. This statistic offers us, beyond the sample size, the power of the results obtained. However, to avoid confusion for the reader, and as indicated by the reviewer, we have redrafted the Discussion section concentrating the discussion on the results of significance.
-
How many repetitions of knee extensions had the participants visualised in this experimental research? Information regarding the cadence per minute and the number of movements displayed in total has been included in the Intervention
Round 2
Reviewer 1 Report
I was interested to see the potential of VAR treatment for knee joints in hemophilia patients. I have a few comments on the manuscript.
I think it is necessary to correct the description about the title, consistency with the text, effect size and sample size.
・Is the expression of the cohort study appropriate? Although it is a prospective intervention study, it has not been followed up over time. I think it needs to be changed.
・Line89 「The aim of this pilot study was to assess~」
Is the position of this research a pilot study?
・Line109 「Assuming a mean effect size (f = 0.30), with an alpha level (type I error) of 0.05 and a statistical power of 80% (1-β = 0.80), a sample size of 13 patients with hemophilic knee arthropathy was estimated.」
In power analysis, the minimum number of participants required for a corresponding difference test (significance level 0.05, power 0.8, effect size 0.3) is 90. For an effect size of 0.89, the minimum number of participants is 12. The wording in the method is inappropriate. Is it pre or post power analysis?
・Line261 「The aim of this cohort study~」
Is there any other appropriate word to describe the design of this study other than the cohort study?
・Line307 「The sample size calculation is for a small effect size (0.30).」
In relation to line109, the sentence meaning is unclear. Significant differences are affected by sample size. For effect size, even if it is not significant, the effect size can be large. Therefore, the effect size needs to be discussed.
・Line338~ Conclusions
The Abstract is positive. The Conclusions do not state that Immersive VR visualization of knee extension movements has the potential to activate the rectus femoris muscle based on the results of the large effect size. The author's argument lacks consistency.
Author Response
Review Report (Reviewer 1)
I was interested to see the potential of VAR treatment for knee joints in hemophilia patients. I have a few comments on the manuscript.
- I think it is necessary to correct the description about the title, consistency with the text, effect size and sample size. The title of the article has been changed (on the recommendation of one of the reviewers), sections such as the Discussion have been rewritten, the error in the exposition of the sample size calculation has been corrected, and the justification for the effect size value has been included.
- Is the expression of the cohort study appropriate? Although it is a prospective intervention study, it has not been followed up over time. I think it needs to be changed. The study design is prospective and observational. Although the non-observational cohort studies and in the intervention, there were two evaluations and a factor that can cause changes (VAR), it has been included in the text that the study is observational.
- Line89. “The aim of this pilot study was to assess…” Is the position of this research a pilot study? The study design is prospective and observational.
- Line109. “Assuming a mean effect size (f = 0.30), with an alpha level (type I error) of 0.05 and a statistical power of 80% (1-β = 0.80), a sample size of 13 patients with hemophilic knee arthropathy was estimated”. In power analysis, the minimum number of participants required for a corresponding difference test (significance level 0.05, power 0.8, effect size 0.3) is 90. For an effect size of 0.89, the minimum number of participants is 12. The wording in the method is inappropriate. Is it pre or post power analysis? Reviewing the analysis of the sample size calculation we have realized the error in the transcription of the text and the translation. Indeed, as the reviewer indicates, if the effect size were so low, a much larger sample size would be necessary. The numerical data in the text have been corrected, according to the analysis of the sample size that was carried out prior to the recruitment of the patients.
- Line261 “The aim of this cohort study…” Is there any other appropriate word to describe the design of this study other than the cohort study? The study design is prospective and observational. Although the non-observational cohort studies and in the intervention, there were two evaluations and a factor that can cause changes (VAR), it has been included in the text that the study is observational.
- Line307. “The sample size calculation is for a small effect size (0.30)”. In relation to line109, the sentence meaning is unclear. Significant differences are affected by sample size. For effect size, even if it is not significant, the effect size can be large. Therefore, the effect size needs to be discussed. In the previous review of the text, another reviewer requested to eliminate the data on the effect size as there were no significant differences. According to you, the significance is greatly affected by the sample size and the value of the effect size can guide us in a more specific way about the true power of the results. We have added the exposure of the effect size results in the text.
- Line338. The Conclusions do not state that Immersive VR visualization of knee extension movements has the potential to activate the rectus femoris muscle based on the results of the large effect size. The author's argument lacks consistency. In the previous review, we were asked to eliminate the discussion and conclusions with the effect size values and therefore all references to this statistic were eliminated. As the reviewer points out, effect size values are important, and we have added the phrase in the study conclusions.
Reviewer 2 Report
The authors should take into consideration the italic text formatting or the red text from the pdf file.
Unfortunately, I could not find any rebuttal letter which point out the requested adjustments performed by the authors. I kindly request in the next response that the authors should provide the paper lines where they performed the suggested adjustments.
Major concerns
First review
- The authors should reconsider their paper because pilot studies have other outcomes and goals. Perhaps the authors should consider if this is a pilot study how the research protocol fitted for a larger randomized controlled trial. The pilot study is also used to identify the validity of a further randomized controlled trial. In accordance with the comments of the reviewer, the study design has been reformulated to avoid unnecessary confusion.
Second Round: Although the authors changed the paper title, from "pilot trial" into "cohort", the research and the article does not fit with the cohort studies.
I think the title should be changed to something like "One session effects of Knee motion visualization using immersive virtual reality", and the authors should consider removing the idea of pilot or cohort study.
First Review
- The sample size calculation is for a small effect size (0.30), also, the research was performed on only one session of VR immersive visualisation. These two are major limitations of the research. The authors should emphasize these aspects in the limitations section. Or discuss them regarding the pilot study goals. According to the author, the main limitation of the study is the size of the sample. This aspect has been included in the study limitations section for quick identification. In the same way, the need to perform RCTs that show the results of VR immersive visualization in patients with hemophilia has been included.
Second Round: Regarding sample size, the authors must specify what kind of family test and statistical used within G*Power. (For example- regarding T-test- differences in means in matched pair, with the data provided in the paper, 71 subjects are needed). Considering the authors' statement in lines 200-201 "Changes after the experimental test were obtained through a paired samples t-test.", the estimated sample size must be wrong. Also, the authors should state if the sample size was calculated before or after the research was performed.
First review
- No data regarding the muscle strength of the quadriceps muscles, although a dynamometer was also used in the assessment. Could the strength interfere with motor neuron fiber recrutation and upper nervous system plasticity response? Also, the authors should emphasize that neuroplasticity is developed in time, 15 minutes are not enough for the conclusions presented. The aim of using a dynamometer was to always request a maximum contraction of 5 seconds from the patient. The dynamometer with an auditory stimulus at 5 seconds facilitates its performance for the patient. In this case the force data were not recorded because due to the mechanism of action in no case would we have adaptations in force. Strength is one of the characteristics of muscle contraction, but in this case, how that muscles are activated is evaluated. If we go to the other extreme and think of high-level sports subjects with great strength development, we can find subjects that do not have a good activation pattern. We couldn't agree more with the reviewer that neuroplasticity develops over time. Therefore, these changes were not expected to be maintained over time if no more sessions were carried out since learning (in this case neuromotor) requires time. However, this work indicates the short-term effect of neuronal activation caused by seeing a movement; how you could improve that recruiting almost immediately. This finding could be very useful clinically to improve motor control in patients with hemophilic arthropathy before exposing them to an active phase.
Second round: The authors should not provide explanation to the reviewer. The information needs to be found in the paper- I could not find any new information regarding the observation made in the first review. This is a scientific journal and is not the case to “go to the other extreme and think of high-level sports subjects with great strength development, we can find subjects that do not have a good activation pattern.” My question was related to the scientific purpose and if the authors could register also the muscle strength developed by the subject quadriceps muscles, perhaps correlations or differences could be found. Therefore, I think the authors should provide further data in this aspect, as was addressed in the introduction section (lines 49-52).
Second round review
Lines 278-281: I don’t think the authors understand the impact and the mechanism of action on human brain regarding electrical transcranial stimulation. The statement that they stimulated the brain using immersive VR for 15 minutes is not based on any objective method of assessment (like MRI). Again I underscore the importance of discussing neuroplasticity and further muscle activation.
The discussion section needs improvement like fluency and consistency.
Author Response
Review Report (Reviewer 2)
The authors should take into consideration the italic text formatting or the red text from the pdf file. Unfortunately, I could not find any rebuttal letter which point out the requested adjustments performed by the authors. I kindly request in the next response that the authors should provide the paper lines where they performed the suggested adjustments.
Major concerns
- Second Round: Although the authors changed the paper title, from "pilot trial" into "cohort", the research and the article does not fit with the cohort studies. I think the title should be changed to something like "One session effects of Knee motion visualization using immersive virtual reality", and the authors should consider removing the idea of pilot or cohort study. As recommended by the author, we have modified the title of the article: One session effects of knee motion visualization using immersive virtual reality in patients with hemophilic arthropathy.
- Second Round: Regarding sample size, the authors must specify what kind of family test and statistical used within G*Power. (For example- regarding T-test- differences in means in matched pair, with the data provided in the paper, 71 subjects are needed). Considering the authors' statement in lines 200-201 "Changes after the experimental test were obtained through a paired samples t-test.", the estimated sample size must be wrong. Also, the authors should state if the sample size was calculated before or after the research was performed. Reviewing the analysis of the sample size calculation we have realized the error in the transcription of the text and the translation. Indeed, as the reviewer indicates, if the effect size were so low, a much larger sample size would be necessary (for an effect size of 0.80, the minimum number of participants is 12). The numerical data in the text have been corrected, according to the analysis of the sample size that was carried out prior to the recruitment of the patients.
- Second round: The authors should not provide explanation to the reviewer. The information needs to be found in the paper- I could not find any new information regarding the observation made in the first review. This is a scientific journal and is not the case to “go to the other extreme and think of high-level sports subjects with great strength development, we can find subjects that do not have a good activation pattern.” My question was related to the scientific purpose and if the authors could register also the muscle strength developed by the subject quadriceps muscles, perhaps correlations or differences could be found. Therefore, I think the authors should provide further data in this aspect, as was addressed in the introduction section (lines 49-52). First, excuse me, but we understood that you were requesting a presentation of the subject. We agree with his comment regarding the (necessary) assessment of muscle strength in patients undergoing a VAR protocol. As this variable was not evaluated in our study, we have included it in the Recommendations for future research
- Second round review: Lines 278-281: I don’t think the authors understand the impact and the mechanism of action on human brain regarding electrical transcranial stimulation. The statement that they stimulated the brain using immersive VR for 15 minutes is not based on any objective method of assessment (like MRI). Again I underscore the importance of discussing neuroplasticity and further muscle activation. To avoid confusion for the reader, we have removed references to impact and the mechanism of action on human brain regarding electrical transcranial stimulation. A discussion focused on neuroplasticity and increased muscle activation has been included.
- The discussion section needs improvement like fluency and consistency. A major rewrite of the Discussion section has been made